# A DNA barcode library of Austrian geometridae (Lepidoptera) reveals high potential for DNA-based species identification

Benjamin Schattanek-Wiesmair[1,2], Peter Huemer[1], Christian Wieser[3], Wolfgang Stark[4], Axel Hausmann[5], Stephan Koblmüller[2], Kristina M. Sefc[2]*

**1** Tiroler Landesmuseen Betriebsges.m.b.H., Innsbruck, Austria, **2** Institute of Biology, University of Graz, Universitätsplatz, Graz, Austria, **3** Landesmuseum Kärnten, Klagenfurt am Wörthersee, Austria, **4** Ökoplus Umweltforschung und Consulting GmbH, Trübensee, Austria, **5** Zoologische Staatssammlung München, München, Germany

* kristina.sefc@uni-graz.at

**Data Availability Statement:** All specimen data and images are publicly available in the dataset

## Abstract

Situated in the Eastern section of the European Alps, Austria encompasses a great diversity of different habitat types, ranging from alpine to lowland Pannonian ecosystems, and a correspondingly high level of species diversity, some of which has been addressed in various DNA barcoding projects. Here, we report a DNA barcode library of all the 476 species of Geometridae (Lepidoptera) that have been recorded in Austria. As far as possible, species were sampled from different Austrian regions in order to capture intraspecific genetic variation. In total, 2500 DNA barcode sequences, representing 438 species, were generated in this study. For complete coverage of Austrian geometrid species in the subsequent analyses, the dataset was supplemented with DNA barcodes from specimens of non-Austrian origin. Species delimitations by ASAP, BIN and bPTP methods yielded 465, 510 and 948 molecular operational taxonomic units, respectively. Congruency of BIN and ASAP partitions with morphospecies assignments was reasonably high (85% of morphospecies in unique partitions), whereas bPTP appeared to overestimate the number of taxonomic units. The study furthermore identified taxonomically relevant cases of morphospecies splitting and sharing in the molecular partitions. We conclude that DNA barcoding and sequence analysis revealed a high potential for accurate DNA-based identification of the Austrian Geometridae species. Additionally, the study provides an updated checklist of the geometrid moths of Austria.

## Introduction

Austria is a landlocked Central European country in the intersection of three biogeographic regions [1]. Situated in the Eastern section of the European Alps, a large part of the country's approximately 84,000 square kilometers is assigned to the alpine biogeographic region, with elevations ranging up to 3798 MSL in the Großglockner mountain massive. The lowlands belong to the continental region, and Pannonian influences are evident in the landscape, fauna

"DS-LEATGEOM Lepidoptera (Geometridae) of Austria" (dx.doi.org/10.5883/DS-LEATGEOM) in the Barcode of Life Data Systems (BOLD, www.boldsystems.org). DNA sequences are also deposited in Genbank (see the Supporting Information for the accession numbers).

**Funding:** Funding for DNA sequencing was provided by the following institutions: - Landesmuseum Kärnten CW, https://landesmuseum.ktn.gv.at/ - Landessammlungen Niederösterreich WS, https://www.landessammlungen-noe.at - Tiroler Landesmuseen PH, BSW https://www.tiroler-landesmuseen.at/ - Promotion of Educational Policies, University and Research Department of the Autonomous Province of Bolzano - South Tyrol PH, https://errin.eu/members/autonomous-province-bolzanobozen-south-tyrol Funding for Open Access publication was provided by University of Graz.

**Competing interests:** The authors have declared that no competing interests exist.

and flora of the northeastern part of the country, where the lowest point of Austria lies at 114 MSL. On a more regional scale, heterogeneity in geological, geomorphological and climatic characteristics is reflected in the classification of eight distinct ecological regions within Austria ([2]; Fig 1).

Sampling sites of Austrian specimens are marked by red dots, and red lines delimit the three sampling areas (North-Eastern Austria, Southern Austria, Western Austria). The map was drawn using the following data: Hillshades, source HeiGIT, used with permission from HeiGIT (Heidelberg Institute for Geoinformation Technology). Administrative borders, source Eurostat (https://ec.europa.eu/eurostat/web/gisco/geodata/reference-data/administrative-units-statistical-units/nuts), download date: February 15, 2023, copyright information: © EuroGeographics, © TurkStat. Source: European Commission–Eurostat/GISCO. Ecoregions, modified from GIS data (https://www.data.gv.at/katalog/dataset/0cf499e1-ac26-47d6-8f8f-afe03d0cc5c7) provided by Umweltbundesamt GmbH under a CC BY 4.0 license (https://creativecommons.org/licenses/by/4.0/).

The exceptionally high ecosystem diversity of Austria is mirrored in high levels of species diversity, with hotpots of regional endemism in the alpine region [3]. For instance, with more than 4070 species of butterflies and moths [4], the lepidopteran species richness of Austria exceeds that of all other Central European countries [5]. In total, over 54,000 animal species have been reported to occur in Austria [6]. However, while species counts have been on the rise in recent decades (in part due to intensified research efforts), the decline of overall biomass and the increasing number of endangered species testify to the pressure experienced by the biodiversity of this biogeographically richly structured area [6]. The stock-taking of Austrian species has recently gained momentum through various DNA barcoding projects, covering a wide range of animal taxa from fish [7], amphibians and reptiles [8], acanthocephalans (e.g.

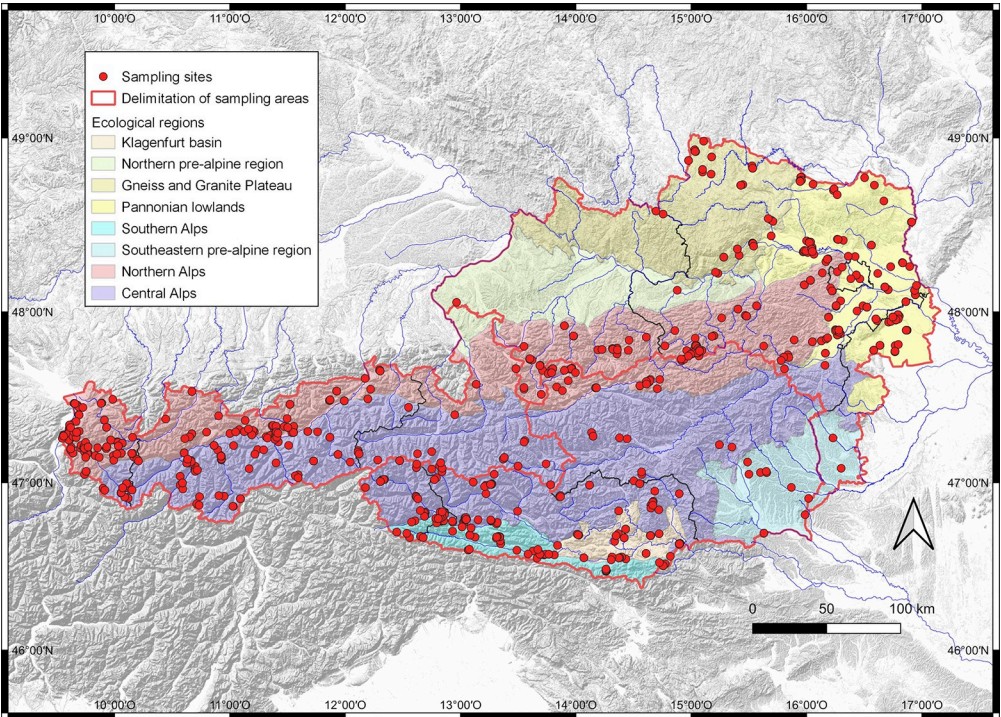

**Fig 1. Map of sampling locations.**

[9, 10]), proturans [11, 12] to various insect taxa (e.g. Odonata:[13]; Orthoptera: [14]; Boreidae: [15]; Ceratopogonidae: [16]; mosquitos: [17, 18]. With respect to Lepidoptera, DNA barcoding has been almost completed for Austrian butterfly species (superfamily Papilionoidea, [19]) and noctuid moths (superfamily Noctuoidea, [20]), and barcoding of the European species of microlepidopteran Gelechiidae and the leaf mining Gracillaridae included most of the species recorded from Austria [21, 22]. Moreover, the lepidopteran fauna of individual federal states has been comprehensively covered [23, 24].

A highly diverse family of Lepidoptera are the Geometridae with approx. 24,000 species described worldwide [25–27] and divided in eight [28, 29] or nine subfamilies [26]. While the largest diversity of geometrid moths is found in tropical regions, six or seven of the subfamilies occur in Europe (Larentiinae, Geometrinae, Ennominae. Sterrhinae, Orthostixinae, Desmobathrinae and Archiearinae; [30], with Orthostixinae possibly requiring integration in Desmobathrinae [26]. Over two decades, the extensive taxonomic research on the approximately 1,000 European species of geometrid moths has been compiled in the book series "The Geometrid Moths of Europe" [31–36]. Recently, the taxonomic work on geometrid moths has been complemented by DNA barcoding efforts [37, 38], occasionally giving rise to the discovery of new species [32, 39–42]. Almost half of the European geometrid species are also found in Austria. The most recent checklist of the Austrian Lepidoptera fauna lists 473 Geometridae species [4], and several small taxonomic changes [32, 43–45] have since then raised the national species count to 476 (see updated checklist of Austrian geometrid moths in S1 Table). The biological and morphological diversity within geometrid moths includes the evolution of behavioural strategies such as day activity as well as morphological adaptations to cold and windy climate conditions reflected in winglessness or brachyptery in the females of some species. These adaptions allow Geometridae to occupy nearly all Austrian habitat types from the lowlands to the subnival zone. In particular, structured and fragmented alpine habitats likely promote diversification and regional endemism. Two endemic species, *Elophos zirbitzensis* (Pieszczek, 1902) and *Sciadia innuptaria* (Herrich-Schäffer, 1852), as well as several subendemic species and endemic subspecies occur in the Austrian alpine region [3, 4]. These and other alpine species and subspecies are restricted to alpine habitats and threatened by climate warming [46]. A national red list for geometrid moths is lacking, but red lists exist for some of the Austrian provinces and report alarmingly large numbers of endangered species, including 139 endangered species in Vorarlberg (corresponding to 38.9% of geometrid species known in the province) [47], 132 species endangered in Salzburg [48], 139 (36.0%) species in Upper Austria [49] and 196 (49.5%) species in Carinthia [50]. Furthermore, two species listed in annex 2 and 4 of the European Habitat Directive occur in the Eastern Austrian provinces Lower Austria and Burgenland (*Chondrosoma fiduciaria* Anker, 1854 and *Lignyoptera fumidaria* (Hübner, 1825)).

The present study assembled DNA barcodes for all 476 Austrian geometrid species. A particular effort was made to sample with broad geographic coverage in order to capture intraspecific genetic variation. We use these data to evaluate the potential for DNA-based species identification, which is crucial, for instance, to meta-barcoding applications for monitoring purposes [51]. We also discuss causes and potential taxonomic consequences of incongruencies between current classification and genetic patterns (BIN-sharing, BIN-splitting).

## Material and methods

### Sampling

In this study, we compiled a dataset comprising DNA barcode sequences of all of the 476 species of Geometridae occurring in Austria. Of these, 438 species were represented by at least

one specimen collected in Austrian, while the remaining 38 species were represented by specimens of non-Austrian origin (see details below). Specimens were collected during faunistic research or were acquired from existing collections. Twenty-eight research collections contributed to this study, with most specimens originating from Tiroler Landesmuseum Ferdinandeum (Innsbruck) (1333 specimens), the research collection of Wolfgang Stark (474 specimens), Landesmuseum Kärnten (Klagenfurt) (376 specimens) and inatura (Dornbirn) (142 specimens). Our sampling was designed with the aim to maximise geographic coverage. Therefore, we divided Austria in three areas: (1) North-Eastern Austria, encompassing the provinces Burgenland, Vienna, Lower Austria and Upper Austria, and including much of the northern pre-alpine region, the Pannonian lowlands and the Gneiss and Granite Plateau; (2) Southern Austria (Styria, Carinthia and East Tyrol), consisting of alpine and pre-alpine regions; and (3) Western Austria (Salzburg, North Tyrol, Vorarlberg), mostly belonging to alpine ecological region. Sample coverage of the three areas was achieved for 272 species, and 79 species were sampled from two areas. The remaining species either occur in only one of the areas [4], or sample collection from the other areas failed although the species has been recorded there. In any case, we aimed at sampling at least three specimens per species. However, for 38 species reported to occur in Austria, no DNA barcode sequences could be generated due to lack of Austrian samples or insufficient quality for successful sequencing. In order to represent these species in the analyses, we retrieved one COI sequence per species from the Barcode of Life Data System (BOLD; www.boldsystems.org [52]), selecting database entries from the geographically nearest sampling sites (mostly from Germany, but ranging from the Iberian Peninsula to Finland; S1 Table).

## Specimen identification

Identification of the morphospecies was based on the "Geometrid moths Europe" series [31–36]. In species where identification based on external morphology was not possible, a dissection of the genitalia was performed. After examination, genitalia were stored in glycerine in a vial pinned underneath the specimen. For comparisons of genitalia morphology between specimens assigned to different BINs within *Gagitodes sagittata* and within *Tephronia sepiaria*, genitalia were fixed on slides following the method of [53]. Photographs of the genitalia preparations were taken with a Panasonic Lumix GH4 mounted on an Olympus BH-2 microscope. The pictures were stacked with Helicon Focus 8 software (HeliconSoft, Ukraine). Adobe Photoshop CS6 and GIMP 2.10.8 (https://www.gimp.org/) were used to add scale bars, assemble photographs of female genitalia and remove background noise.

## DNA barcoding and data analysis

Dry legs of 2734 specimens were sent to the Canadian Centre for DNA Barcoding (CCDB, Biodiversity Institute of Ontario, University of Guelph) for sequencing the DNA barcode region of the mitochondrial COI gene (cytochrome c oxidase subunit 1). The DNA barcode sequences were generated using a standard high-throughput protocol [54] using primers LepF1 and LepF2 [55]. Resulting sequences were checked for DNA barcode compliance in the BOLD system, and sequences > 500 bp that met these criteria were retained for further analyses. Specimen collection data and images are publicly available in the BOLD dataset "DS-LEATGEOM Lepidoptera (Geometridae) of Austria", and DNA sequences were also deposited in Genbank (Genbank accession numbers in S1 Text).

## Data analysis

For each species, the nearest neighbor Kimura two-parameter (K2P) distances (min.NN) as well as mean and maximum intraspecific K2P distances (for species with > 1 sample) were

calculated on the BOLD system v. 4.0 using the Barcode Gap Analysis tool, with pairwise deletion of missing/ambiguous characters. Nearest neighbor distances were calculated between the focal species and the most similar COI sequence of the nearest neighbor species in the dataset.

Sequences were assigned Barcode Index Numbers (BINs; [56]). BINs are based on an algorithm that clusters all high-quality sequences from BOLD into operational taxonomic units, regardless of their previous taxonomic assignment. We recorded the number of BINs per species, that were detected in the Austrian specimens. For each of the 476 Austrian geometrid species, we also determined the number of BINs per species in datasets including all European specimens (excluding Russian and Turkish specimens). These data were derived from BOLD in March 2022 and were used to plot the number of intra-specific BINs detected in Austria against the number of intra-specific BINs of the same species present across Europe.

Additionally, molecular operational taxonomic unit (MOTU) delimitation was carried out with ASAP [57] and bPTP [58]. The assignment of species to subfamilies follows the systematic checklist of European Geometridae [30]. ASAP is a distance-based method to partition sequence datasets into MOTUs and was calculated on the webserver (https://bioinfo.mnhn.fr/abi/public/asap/) using the K2P model with ts/tv set to 2. Finally, we used a Bayesian implementation of the PTP model for species delimitation (bPTP), which is based on a rooted phylogenetic tree and predicts branching events and speciation based on the numbers of substitutions. The maximum likelihood (ML) trees to be used as input files for the bPTP analysis were constructed via the Phylosuite v.1.2.2 platform [59]. The following plugins were used: The alignment was performed with MAFFT v7.313 [60], the best fitting model was calculated with Modelfinder (IQ-TREE v.1.6.8) [61] and the maximum likelihood tree was inferred with IQ-TREE v.1.6.8 [62] with the setup of an automatic substitution model and 20000 ultrafast bootstrap replicates.

In addition to the output retrieved from BOLD, summary statistics were calculated and plotted in R v. 4.1.3 (R Development Core Team 2022). The bPTP analysis split numerous species into several MOTUs, and the relationship between numbers of bPTP-MOTUs per species and sample size was examined in a generalized linear model with a negative binomial error distribution using the R package glmmTMB [63].

Phylogenetic relationships within individual taxonomic groups were represented by Neighbor Joining trees based on K2P distances (with pairwise deletion of missing/ambiguous characters) constructed using MEGA X [64]. The trees were visualized with FigTree v1.4.4 [65] and GIMP 2.10.8 [66].

## Results

### DNA barcode sequence generation

In this study, 2567 sequences (>500 bp) of specimens of Austrian origin were generated, covering 438 species (92% of the Austrian geometrid species). Of these, 2500 sequences fulfilled the criteria of DNA barcode compliance and 2525 sequences were assigned a Barcode Index Number (BIN). Sequencing failed entirely for 118 specimens, and 49 sequences of lengths < 500 bp were excluded from further analyses.

Across all species (n = 476, including the 38 species that were represented by non-Austrian specimens), sample sizes ranged from 1–24 specimens per species (median = 5.0). For 365 species, we achieved our goal to obtain three or more DNA barcode sequences, while 44 species were represented by two DNA barcode sequences and 67 species were represented by a single DNA barcode sequence.

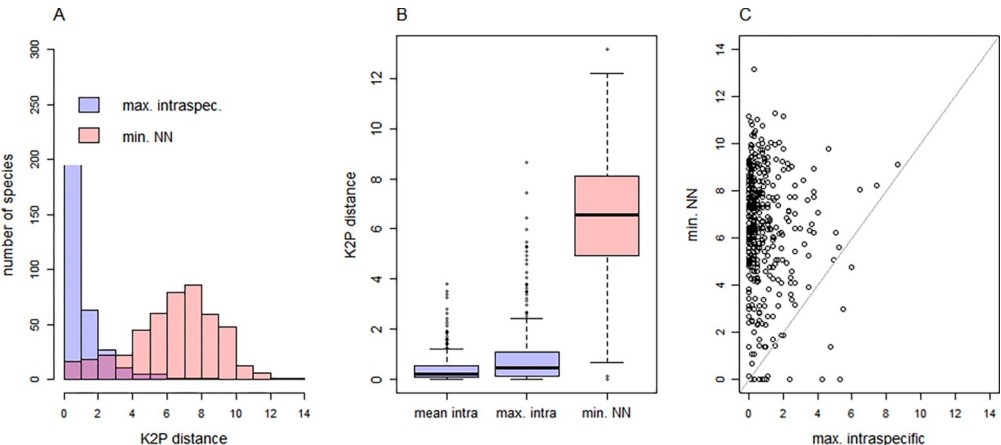

**Fig 2. Intraspecific distances and distances to nearest neighbor.** (A) Histograms of maximum intraspecific distance (max. intraspec.) and minimum distances to the nearest species (min.NN) for the investigated species. (B) Boxplots of mean intraspecific distances (mean intra), maximum intraspecific distances (max. intra) and minimum distances to nearest neighbor species (min. NN). (C) Scatterplot of maximum intraspecific against minimum interspecific (min. NN) distances illustrating the barcoding gap for species plotted above the diagonal line. All distances were calculated using the K2P model.

### Intra- and interspecific genetic distances

In species with two or more DNA barcode sequences (n = 409 species), mean intraspecific K2P distances ranged from 0% to 3.81% (average across species = 0.42; median = 0.21%), and maximum intraspecific K2P distances ranged from 0% to 8.7% (average across species = 0.88%; median = 0.46%; Fig 2A and 2B).

Across all species (n = 476 species), minimum interspecific distances to the nearest neighbor (min.NN) ranged from 0% to 13.15% (average across species = 6.37%; median = 6.57%; Fig 2A and 2B), and fell below 2% for 34 species (7.1% of 476 species; S1 Table).

For the vast majority of species (97.5%), min.NN exceeded their maximum intraspecific distance (Fig 2C).

### Species delimitation

Species delimitation by means of ASAP, the BIN system and bPTP yielded 465, 510 and 948 molecular operational taxonomic units (MOTUs) across the full dataset covering 476 morphospecies. BIN and ASAP yielded identical MOTU assignments for 417 morphospecies, 269 of which were also retrieved by bPTP (Fig 3). In contrast, for 166 species, their bPTP-based MOTU assignments were not supported by the other algorithms (Fig 3).

Congruency between morphospecies and BIN assignment was high, with unique BINs assigned to 406 (85.3%) of the 476 morphospecies. 27 morphospecies shared BINs with another morphospecies (see Table 1), 47 morphospecies were split into two BINs, and two morphospecies (*Eupithecia icterata* and *Eupithecia satyrata*) were split into three BINs (Fig 4). Four morphospecies showed both BIN-splitting and BIN-sharing (*Elophos caelibaria*, *Rheumaptera subhastata*, *Sciadia tenebraria*, *Sciadia zelleraria*; S1 Table). Broken down by subfamily (Table 2, S2 Table), congruence between BINs and morphospecies assignments was weaker in the species-poor subfamilies Geometrinae and Archiearinae than in the other three, species-rich subfamilies.

Across Europe, 140 (i.e., 30%) of the here investigated species split into multiple BINs, with up to six BINs per morphospecies. In Austria, 91 of these species are represented with a single BIN each, and 47 species are each represented with two BINs (Fig 6; S1 Table). Species

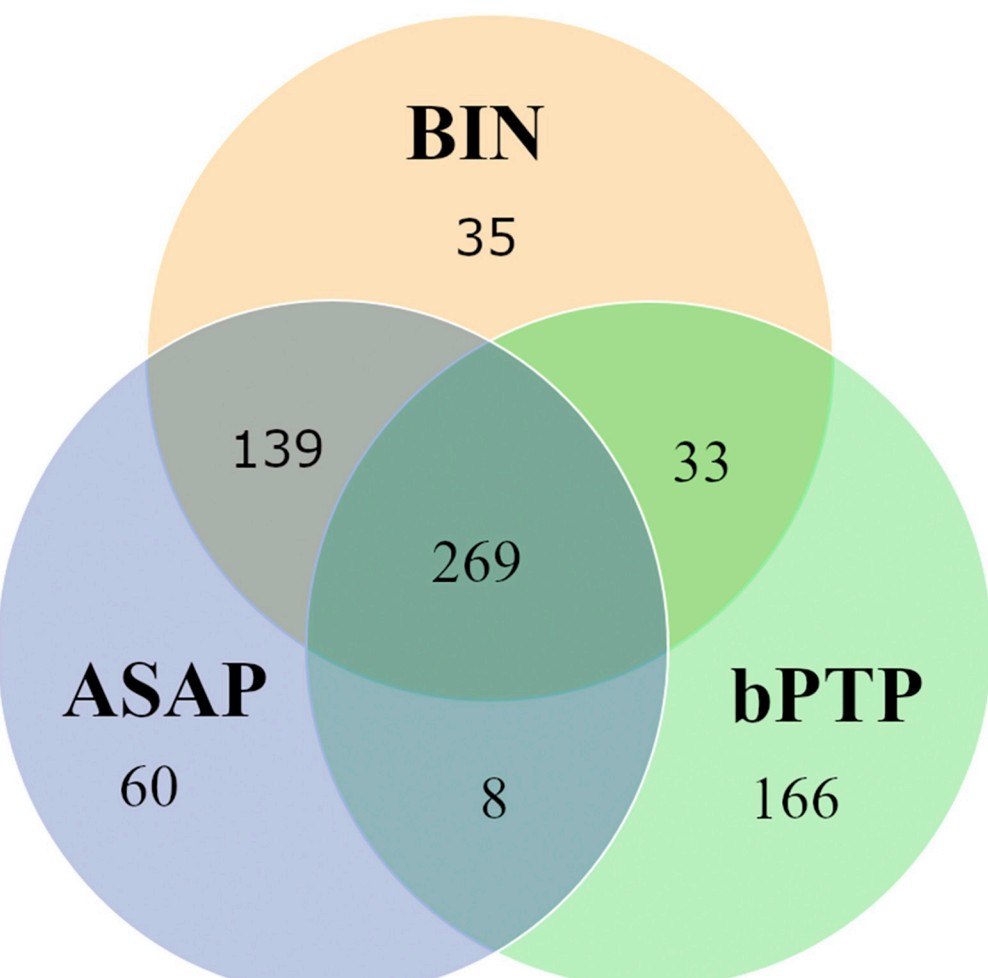

**Fig 3. Venn diagram illustrating congruence and incongruence of MOTU assignment between the three methods used in this study.** Number of species, for which MOTU assignments were consistent (overlapping areas) or inconsistent across species delimitation methods.

subjected to BIN splitting, in which maximum intraspecific distance exceeded 3% [71], are listed in Table 3. Most of these BIN splits were supported by ASAP, and even higher numbers of MOTUs per morphospecies were suggested by bPTP (Table 3).

Based on the ASAP method, 396 morphospecies (83.1% of the Austrian geometrid species) were assigned to unique MOTUs (Fig 4). For a total of 61 morphospecies, ASAP did not discriminate between two or more morphospecies, and 22 morphospecies were split into two MOTUs (Fig 4). Three morphospecies experienced both MOTU splitting and MOTU sharing (*Rheumaptera subhastata*, *Sciadia tenebraria*, *Sciadia zelleraria*; S1 Table). Differences from BIN results concerned mainly the species-poor subfamilies (Geometrinae and Archiearinae), where ASAP achieved higher congruence with morphospecies delimitation than did the BIN method (Table 2). Threshold distances for MOTU-partitioning varied among subfamilies from K2P = 1.2% to K2P = 5.3% (Geometrinae, 5.3%; Archiearinae, 1.2%; Ennominae, 3.0%; Larentiinae, 4.1%; Sterrhinae, 2.2%).

The bPTP analysis resulted in the discrimination of 948 MOTUs. Only 282 morphospecies (59.2%) were assigned to individual unique MOTUs, while 181 morphospecies (36.6%) were

**Table 1. BIN-sharing between morphospecies.**

| Species (sample size) | Min. NN (K2P) | Samples cluster by species in phylogenetic tree | MOTU shared | | Differentiation | Comments |
|---|---|---|---|---|---|---|
| | | | ASAP | bPTP | | |
| *Lycia alpina* (n = 5) *Lycia zonaria* (n = 3) | 0 | no | yes | yes | wing coloration and wing shape, habitat preferences | Hybridization reported [32,67]. The same BIN is shared also by another (non-Austrian) species, *L. graecarius*. |
| *Lycia isabellae* (n = 1) *Lycia pomonaria* (n = 1) | 1.26 | no[a] | yes[b] | no | wing coloration and wing shape, larval host plants, habitat preferences | Hybridization reported [32,67] |
| *Perizoma affinitata* (n = 6) *Perizoma hydrata* (n = 7) | 0.15 | no | yes[c] | Yes | wing coloration, male and female genitalia | Low interspecific divergence also in [38] |
| *Cyclophora punctaria* (n = 8) *Cyclophora quercimontaria* (n = 1) *Cyclophora suppunctaria* (n = 1) | 0–0.15 | no [a] | yes | yes | male and female genitalia | Hybridization reported. Species barriers are generally low in the genus *Cyclophora* and natural hybrids were found repeatedly [34,68]. |
| *Thera cembrae* (n = 20) *Thera obeliscata* (n = 10) | 0 | no | yes | yes | host plants | *T. cembrae* is differently interpreted: as valid species but possibly conspecific with *T. obeliscata* [36,69]; or as subspecies of *T. variata* [70], which is refuted by the current sequence data ([Fig 5]) |
| *Sciadia zelleraria* (n = 8) *Sciadia tenebraria* (n = 10) *Sciadia innuptaria* (n = 4) | 0 | no | yes | yes | wing coloration and wing shape | Hybridization in the genus *Sciadia* is common [32]. |
| *Elophos caelibaria* (n = 14) *Sciadia slovenica* (n = 1) | 0 | no [a] | yes | yes | wing coloration and wing shape | Hybridization in the genus *Sciadia* is common [32] and may encompass its sister genus *Elophos* |
| *Chlorissa cloraria* (n = 7) *Chlorissa viridata* (n = 4) | 0 | no | yes | yes | ontroversial: differentiation by forewing costa and in male genitalia [35,70], but overlapping traits at least in some geographic regions reported | Potentially conspecific: Morphological distinction unclear, identical DNA barcodes shared across species |
| *Thera britannica* (n = 9) *Thera variata* (n = 9) *Thera vetustata* (n = 7) | 1.0–2.0 | yes | yes | no | wing coloration, shape of male antenna | |
| *Boudinotiana notha* (n = 4) *Boudinotiana puella* (n = 2) | 1.4 | yes | no | yes | wing coloration and wing shape | |
| *Chloroclysta siterata* (n = 7) *Chloroclysta miata* (n = 7) | 1.4 | yes | yes | no | wing coloration and wing shape | |

(*Continued*)

**Table 1.** (Continued)

| Species (sample size) | Min. NN (K2P) | Samples cluster by species in phylogenetic tree | MOTU shared | | Differentiation | Comments |
|---|---|---|---|---|---|---|
| | | | ASAP | bPTP | | |
| *Rheumaptera hastata* (n = 22) *Rheumaptera subhastata* (n = 5) | 1.39 | yes | yes | no | male genitalia | |

The table lists cases of BIN-sharing between morphospecies, along with sample size per species and the minimum Kimura-2-parameter genetic distance between the involved species (Min. NN K2P). Min. NN distances of 0 indicate identical DNA sequences, i.e. cases of DNA barcode sharing between species. We also indicate whether sequences cluster by species in phylogenetic reconstructions, in which cases DNA-based species discrimination is possible despite BIN sharing; and whether MOTU sharing was also observed in ASAP and bPTP analyses. Finally, we report phenotypic differences between BIN-sharing species and, when possible, offer explanations for the observed BIN sharing.

[a] only 1 sample for one or both species.

[b] together with *L. hirtaria*.

[c] together with *Perizoma lugdunaria*.

split into multiple MOTUs and 20 morphospecies (4.2%) shared their MOTU with another species (Fig 4). Seven of these morphospecies showed both MOTU splitting and MOTU sharing (*Chlorissa cloraria*, *Chlorissa viridata*, *Elophos caelibaria*, *Rheumaptera subhastata*, *Sciadia tenebraria*, *Sciadia zelleraria*, *Thera cembrae*, *Thera obeliscata*; S1 Table). Five morphospecies were split into ten or more MOTUs (highest MOTU numbers:18 MOTUs in *Eupithecia virgaureata*, 12 MOTUs in *Charissa supinaria*, 11 MOTUs in *Eupithecia abietaria* and in *Operophtera brumata*, and10 MOTUs in *Eupithecia intricata*). Congruence between MOTUs and morphospecies was similarly poor across all subfamilies (Table 2). We found bPTP partitioning to be positively correlated with sampling effort, as the number of bPTP-MOTUs per species increased significantly with sample size per species (Fig 9; GLM: est. = 0.08, z = 8.78, $p < 10^{-16}$, based on n = 409 species with sample size > 1).

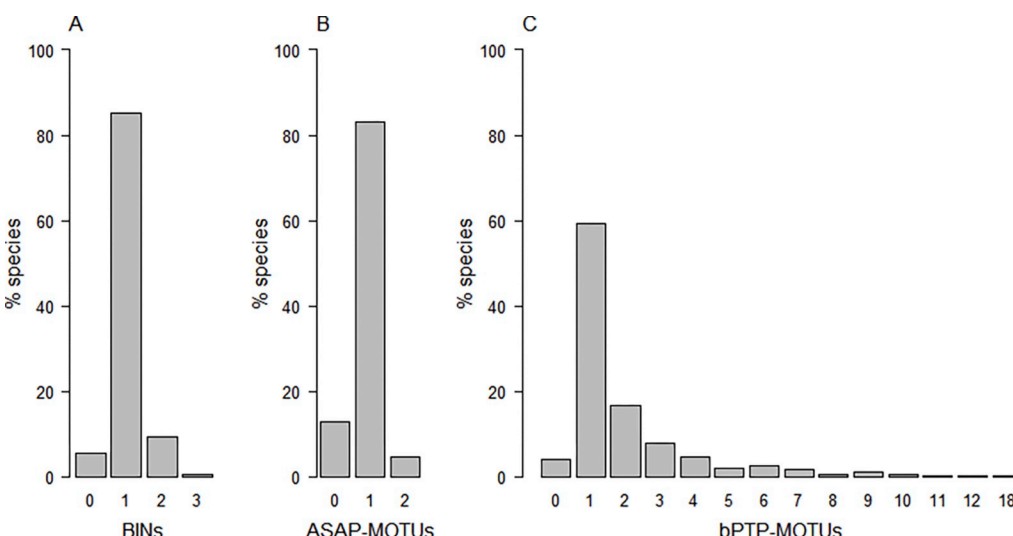

**Fig 4. Congruence between MOTU assignments and morphospecies classification.** The barplots show the percentage of species that share MOTUs with another species (category 0 = MOTU sharing), are assigned to their unique individual MOTU (category 1 = congruence with morphospecies), or are split into multiple MOTUS (categories 2 and higher = MOTU splitting). (A) BIN assignments; (B) MOTUs as defined by ASAP; (C) MOTUs as defined by bPTP.

**Table 2. Morphospecies and MOTUs of Austrian Geometridae.**

| taxonomic group | no. of species | BIN | | | ASAP | | | bPTP | | |
|---|---|---|---|---|---|---|---|---|---|---|
| | | unique | share | split | unique | share | split | unique | share | split |
| Geometridae | 476 | 406 | 27 | 47 | 396 | 61 | 22 | 282 | 20 | 181 |
| By subfamily: | | | | | | | | | | |
| Geometrinae | 13 | 10 | 2 | 1 | 9 | 4 | 0 | 7 | 2 | 6 |
| Archiearinae | 3 | 0 | 2 | 1 | 3 | 0 | 0 | 1 | 2 | 0 |
| Ennominae | 146 | 122 | 9 | 18 | 123 | 16 | 9 | 91 | 7 | 51 |
| Larentiinae | 248 | 216 | 11 | 22 | 201 | 38 | 10 | 141 | 6 | 103 |
| Sterrhinae | 66 | 58 | 3 | 5 | 60 | 3 | 3 | 42 | 3 | 21 |

The table reports the number of Geometridae morphospecies (no. of species) recorded in Austria, and the number of morphospecies assigned unique BINs (unique), the number of morphospecies sharing MOTUs with other morphospecies (share) and the number of morphospecies split into multiple MOTUs (split), for each of the three MOTU delimitation methods. Since both MOTU splitting and sharing occurred in some morphospecies, the sums of the species counts across "unique", "share" and "split" may exceed the number of species.

## Discussion

### MOTU delimitation and morphospecies identification

The diversity of European Macrolepidoptera has been a longstanding research focus and is considered to be thoroughly recorded and characterized [71, 72]. This is also true for the family Geometridae, where the high quality and intensity of taxonomic work [31–36, 41, 42] results in morphospecies classifications that obviously also provide a good picture of the underlying genetic structure and diversity, although certain cases still remain disputable (e.g. species pairs *Chlorissa viridata/cloraria*, *Thera cembrae/obeliscata* and a few others). In our study, approximately 85% of morphospecies were represented by unique BINs and ASAP-MOTUs, while potential cases of cryptic diversity were suggested in 10% (BIN) or 5% (ASAP) of the Austrian species. Together with those of the BIN-sharing species whose sequences nonetheless cluster by species in phylogenetic trees (Table 1), this amounts to a total of 464 out of 476 species (97%) that can be identified by their COI sequence. These results correspond well with those of previous barcoding studies in Geometridae [37, 38], as well as with the success rates of genetic species identification that were achieved in Central European Ensifera (100%; [14]), European Apoidea (99%; [73]), Austrian Noctuoidea (98%; [20]), Northern European tachinid flies (93%; [74]), German Heteroptera (92%; [75]), European Coleoptera (92%; [76]), German Neuroptera (90%; [77]), European Odonata (88%; [13]) and European butterfly species (85% [78]). These successes in DNA-based species identification constitute a promising groundwork for monitoring studies that can for instance employ metabarcoding approaches to efficiently analyze insect diversity from Malaise trap or environmental DNA samples in a high throughput manner [79].

In the present study, BIN and ASAP performed similarly well in terms of congruence between reconstructed MOTUs and morphological taxonomy. Being embedded in the BOLD system, BIN makes use of the enormous amount of data in the database in MOTU construction and delimitation. While this is certainly an asset, the limited possibilities to curate the public data (e.g. by editing incorrect taxonomic notation) or to edit the alignment prior to analyses are undesirable constraints. In the framework of the current "Biodiversity Europe (BGE)" project, however, a comprehensive curation of the reference library on BOLD is aimed at a pan-European level. Our results suggest ASAP as a promising, fast alternative for MOTU delimitation and species identification that can be used with more flexibility in alignment construction. In cases of inconsistency with morphospecies classification, ASAP had a stronger

tendency to morphospecies lumping than BIN, while BIN performed a higher rate of morphospecies splitting than ASAP (Fig 4). Similarly, BIN splits were far more common than BIN merges in a large dataset of Canadian spiders [56]. Between the two methods, their relative merits depend on the goal: While BIN seems to be more sensitive than ASAP to cryptic diversity but is perhaps prone to over-splitting, this does not impair species identification, where MOTU splitting is less problematic than MOTU sharing.

In contrast, the performance of bPTP as a species identification method was rather poor, as the number of MOTUs identified by bPTP was about twice as high as the number of morphospecies in the dataset, and was furthermore sensitive to sample size. Oversplitting by bPTP has also been observed in other datasets across various taxa [80–82].

## Incongruences between MOTUs and morphospecies: Sharing and splitting

Approximately 5% of the Austrian geometrid species cannot be identified based on the BIN system, as they share their BINs with one or two other, typically congeneric species (Table 1). One of the BIN sharing dyads was separated by ASAP. In many cases, the BIN sharing species also share DNA barcodes (i.e., have identical haplotypes), or sequences do not cluster by species in phylogenetic reconstructions (Table 1). However, with two exceptions, the BIN sharing species can be discriminated unambiguously based on morphological traits (see Table 1 for distinguishing traits) and likely represent young but distinct species. In some of these cases, present or past hybridization (genetic introgression) may be responsible for interspecific haplotype sharing (Table 1). In contrast, *Chlorissa cloraria* and *C. viridata* are difficult if not impossible to distinguish morphologically and share identical DNA barcodes, suggesting possible synonymy. Likewise, *Thera cembrae* and *T. obeliscata* are morphologically indistinguishable, although different in host plant use, and share identical DNA barcodes. *Thera cembrae* has been considered as possibly conspecific with *T. obeliscata* [36, 69] or as subspecies of *T. variata* [70]. The genetic data (Fig 5) refute the latter proposition, but are concordant with synonymy of the two species.

Approximately 10% of the morphospecies investigated in this study were split into multiple BINs, with intraspecific K2P distances exceeding 3% in 23 species (Table 3). Some of the detected BIN splits, especially those with large K2P distance, may be taxonomically relevant and need to be tested for support by nuclear genetic data and phenotypic traits. Most of the

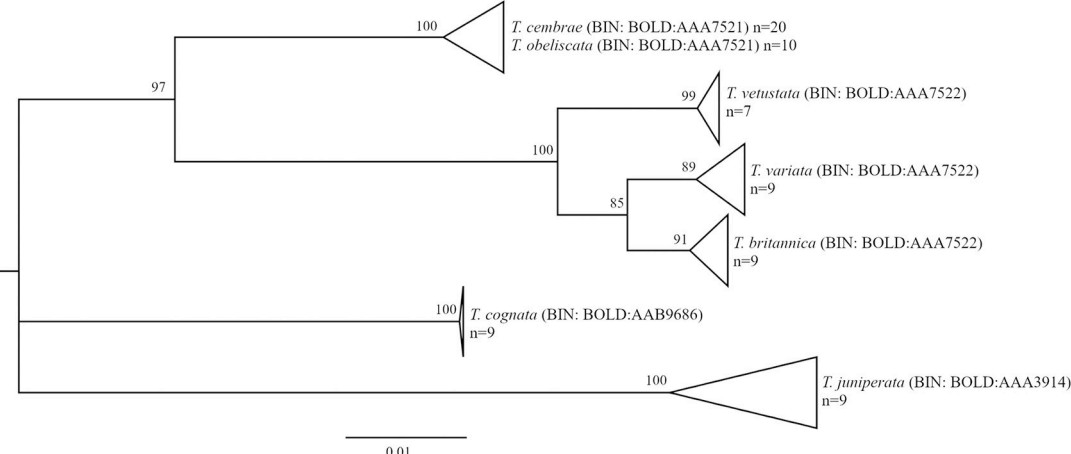

**Fig 5. Phylogenetic relationships among species of the genus *Thera*.** Neighbor Joining trees based on K2P distances (with pairwise deletion of missing/ambiguous characters), n = sample size; all samples are of Austrian origin.

**Table 3. Morphospecies splitting.**

| Species | Sample size | Max. intra. K2P | BINs Austria | Additional BINs in Europe | ASAP-MOTUs | bPTP-MOTUs | Distribution | Comments (e.g., geographic distribution and frequencies of BINs in BOLD database) |
|---|---|---|---|---|---|---|---|---|
| *Electrophaes corylata* | 5 | 8.65 | BOLD: AAZ5334 | | 2 | 2 | overlapping | Both BINs are widespread across Europe. |
| | | | BOLD: AAC3785 | | | | | |
| *Hypomecis punctinalis* | 8 | 7.43 | BOLD: AAB1058 | BOLD: AAB1059 | 2 | 2 | overlapping | High level of DNA polymorphism within BINs [32]. |
| | | | BOLD: ACA2461 | | | | | |
| *Ectropis crepuscularia* | 7 | 6.45 | BOLD: AAA2076 BOLD: ACE6053 | | 2 | 7 | overlapping | Both BINs are widespread across Europe. |
| *Alcis repandata* | 8 | 5.96 | BOLD: AAA8484 BOLD: AAA8482 | | 2 | 3 | overlapping | Both BINs are widespread across Europe. BOLD:AAA8482 is shared with *A. extinctaria* from Altai mountains. |
| *Horisme tersata* | 19 | 5.48 | BOLD: AAC5135 | | 1[a] | 5 | overlapping | BOLD:AAC5135 is widespread across Europe; BOLD:ABW1783 is shared between one Austrian specimen and specimens from Chinese origin |
| | | | BOLD: ABW1783 | | | | | |
| *Sciadia zelleraria* | 19 | 5.31 | BOLD: AAB4957 | BOLD: ACE3560 | 2 | 2 | overlapping | Hybridization among *Sciadia* species is common [32] and may be the origin of heterospecific haplotypes. Note that some haplotypes of *S. zellaria* are shared with *S. tenebraria* and *S. innuptaria* (Table 1). |
| | | | BOLD: AAD2991 | BOLD: ABX0095 | | | | |
| | | | | BOLD: ACJ3674 | | | | |
| *Coenotephria salicata* | 16 | 5.27 | BOLD: AAB9029 BOLD: AAC8889 | | 2 | 9 | overlapping | BOLD:AAB9029 is widely distributed across Europe, BOLD:AAC8889 is restricted to Germany, Austria and England |
| *Pasiphila rectangulata* | 14 | 5.08 | BOLD: AAA3076 BOLD: AAA3075 | | 2 | 5 | overlapping | BOLD:AAA3075 consists of European and Northern American samples, BOLD: AAA3076 only of European samples. |
| *Eupithecia satyrata* | 21 | 4.94 | BOLD: ACT7251 | | 2 | 6 | overlapping | Each of the three BINs is distributed widely across Europe; BOLD:ACT7251 occurs at high frequency. |
| | | | BOLD: AAA4219 | | | | | |
| | | | BOLD: AAA5442 | | | | | |
| *Rheumaptera subhastata* | 7 | 4.75 | BOLD: AAA5436 BOLD: AAA5435 | | 2[b] | 4 | overlapping | BOLD:AAA5435 is more closely related to its sister species *R. hastata* (1.43% K2P) than to the conspecific BIN BOLD: AAA5436. |
| *Gagitodes sagittata* | 3 | 4.62 | BOLD: AAD8985 | | 2 | 3 | overlapping | BOLD:AAD8985 found in Finland, Northern Italy and in Austria (Vorarlberg and Tyrol), BOLD:AAD8984 in Germany (Bavaria) and Austria (Styria; Fig 7). No differences in genital morphology were detected (n = 1 male and 1 female for each BIN). |
| | | | BOLD: AAD8984 | | | | | |

(*Continued*)

**Table 3.** (Continued)

| Species | Sample size | Max. intra. K2P | BINs Austria | Additional BINs in Europe | ASAP-MOTUs | bPTP-MOTUs | Distribution | Comments (e.g., geographic distribution and frequencies of BINs in BOLD database) |
|---|---|---|---|---|---|---|---|---|
| *Sciadia tenebraria* | 10 | 4.27 | BOLD: ACE3562 / BOLD: AAB4957 | BOLD: AAB4956 / BOLD: ACE3563 / BOLD: AAC5403 / BOLD: ACE3561 | 2 | 2 | overlapping | Hybridization among *Sciadia* species is common [32] and may be the origin of heterospecific haplotypes. Note that some haplotypes of *S. tenebraria* are shared with *S. zellaria* and *S. innuptaria* (Table 1). |
| *Eupithecia plumbeolata* | 12 | 4.04 | BOLD: AAB8936 BOLD: AAB8937 | BOLD: ACF3745 | 2 | 2 | overlapping | BOLD:AAB8936, BOLD:AAB 8937 were found across Europe. BOLD:AAB8937 is shared with the Asian species *E. nomogrammata*. BOLD:ACF3745 is known from Finland and Turkey. |
| *Agriopis bajaria* | 5 | 3.81 | BOLD: AEE3670 / BOLD: AAC3211 | BOLD: AAC3209 / BOLD: AAZ7585 | 2 | 5 | non-overlapping | High level of DNA polymorphism within BINs [32] |
| *Elophos operaria* | 3 | 3.80 | BOLD: ADO6177 / BOLD: AEE3881 | | 2 | 2 | non-overlapping | BOLD:ADO6177 represents the nominotypical subspecies; the specimen in BOLD:AEE3881 was collected from the locus typicus (Zirbitzkogel) of the subspecies *E. operaria hoefneri*. [32] found no morphological evidence for subspecies recognition. |
| *Epirrita autumnata* | 10 | 3.75 | BOLD: AAA5906 / BOLD: AAA5907 | BOLD: ACE7803 / BOLD: AAA5909 / BOLD: ABY8748 | 1 | 2 | overlapping | BOLD:AAA5907 is shared with *E. filigrammaria* (an endemite of Great Britain); possibly due to introgression [36]. |
| *Epirrhoe galiata* | 9 | 3.64 | BOLD: ACE4142 / BOLD: ACE4676 | | 1 | 3 | overlapping | Both BINs are widespread across Europe. |
| *Idaea seriata* | 13 | 3.53 | BOLD: AAA9645 BOLD: ACF4900 | BOLD: ABZ4137 BOLD: ABY6334 | 2 | 2 | overlapping | BOLD:AAA9645 is shared with *I. minuscularia* (South-Western Europe). BOLD:ABZ4137 and BOLD:ABY6334 are restricted to southern Italy. |
| *Tephronia sepiaria* | 4 | 3.47 | BOLD: AAD2603 / BOLD: ABV4483 | BOLD: ADK9120 | 2 | 3 | non-overlapping | BOLD:AAD2603 is distributed in Western Europe. BOLD:ABV4483 is shared with specimens from Turkey and Greece (Fig 8); subspecies status of this BIN has been taken into consideration [32]. Minor differences in genital morphology were detected (Fig 10; n = 1 male and 1 female for each BIN). |
| *Eulithis populata* | 6 | 3.47 | BOLD: ADF0720 BOLD: ABZ1837 | | 1 | 2 | overlapping | BOLD:ABZ1837 is widespread and frequent (40 DNA barcode sequences on BOLD); BOLD:ADF0720 currently contains only one Austrian sequence. |
| *Eupithecia subfuscata* | 21 | 3.40 | BOLD: ABY4251 / BOLD: ACE8007 | BOLD: ABY4252 / BOLD: ABW4471 | 1 | 6 | overlapping | BOLD:ABY4251 and BOLD:ACE8007 are widespread across Europe. BOLD: ABY4252, BOLD:ABW4471 are restricted to the Netherlands, with one sequence each. |

(*Continued*)

**Table 3.** (Continued)

| Species | Sample size | Max. intra. K2P | BINs Austria | Additional BINs in Europe | ASAP-MOTUs | bPTP-MOTUs | Distribution | Comments (e.g., geographic distribution and frequencies of BINs in BOLD database) |
|---|---|---|---|---|---|---|---|---|
| *Xanthorhoe spadicearia* | 8 | 3.23 | BOLD: AAB7980 BOLD: AAB7981 | | 1 | 2 | overlapping | BOLD:AAB7980 and BOLD:AAB7981 are widespread across Europe. |
| *Lomaspilis marginata* | 9 | 3.16 | BOLD: AAB5300 BOLD: ABZ2599 | | 1 | 1 | overlapping | BOLD:AAB5300 is frequent and widespread across Europe; BOLD: ABZ2599 has been found Austria and Poland |

The table provides information on morphospecies that were split into multiple BINs with K2P distance > 3%. For each species, sample sizes and maximum intraspecific K2P genetic distances (Max. intra. K2P) are reported, and intraspecific BINs detected in Austria as well as additional BINs detected elsewhere in Europe are identified. Furthermore, we report the number of intraspecific ASAP- and bPTP-MOTUs and indicate whether the distributions of intraspecific BINs overlap geographically. In the last column, we compiled information related to the BIN structure in the species.

[a] together with *H. radicaria*.

[b] one ASAP MOTU shared with *R. hastata*.

intraspecific BINs, which were detected in Austria, are shared with other European specimens of the same species (Table 3). In two cases, Austrian specimens share BINs with geographically more distant specimens: four Austrian samples of *Alcis repandata* are assigned to a BIN

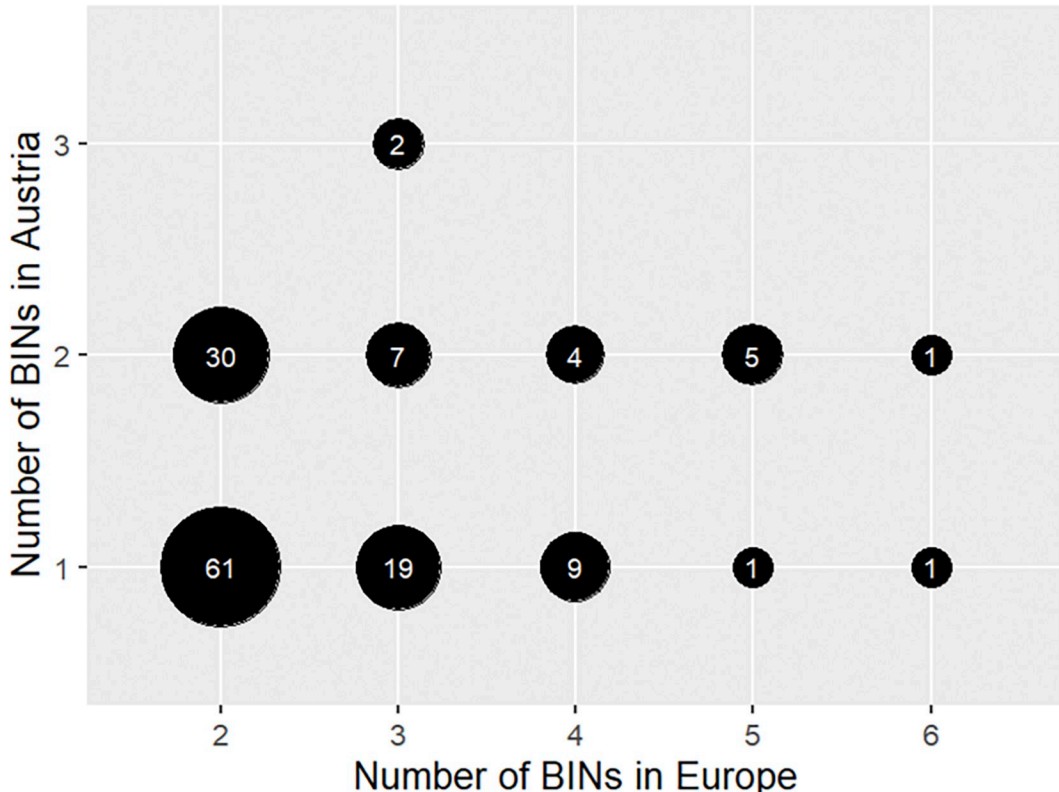

**Fig 6. Number of BINs per species detected in Austria, plotted against the number of BINs in the same species across its European distribution.** Circle size corresponds to the number of species with identical values; the species count is also reported within each circle.

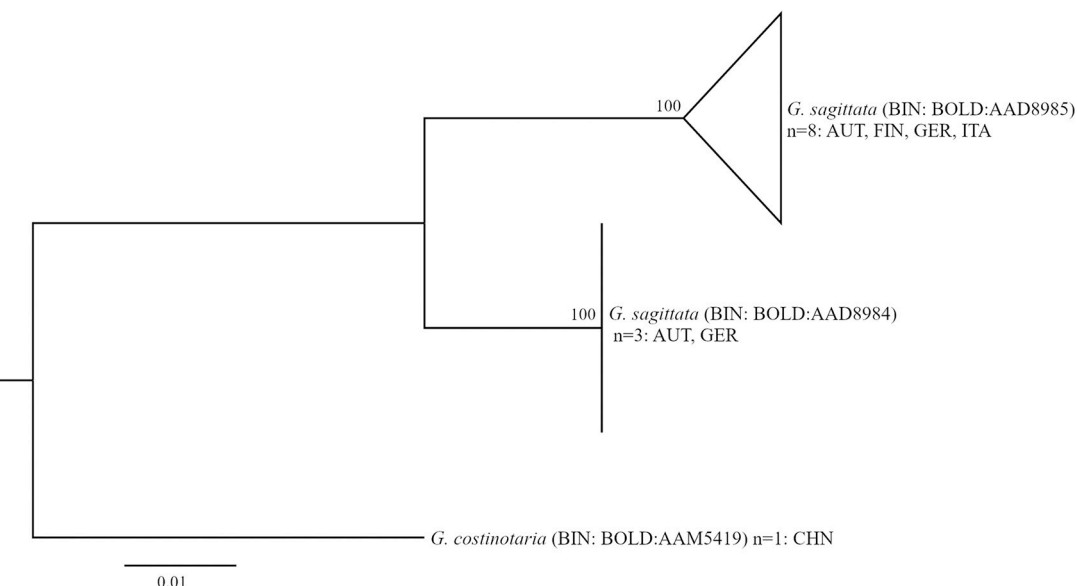

**Fig 7. Phylogenetic relationships among BINs of *Gagitodes sagittata, G. costinotaria*.** Neighbor Joining trees based on K2P distances (with pairwise deletion of missing/ambiguous characters), n = sample size; origin of samples indicated with ISO 3166–1 alpha-3 three-letter country code.

together with conspecific specimens from the Altai mountains [83], and one of the BINs assigned to Austrian *Eupithecia plumbeolata* is shared with the Asian species *E. nomogrammata*.

Furthermore, a new BIN was discovered in the species *Elophos operaria*. This BIN (BOLD: AEE3881) is composed of a single sample that was collected from the type locality of *E. operaria hoefneri* (Rebel, 1903). Müller et al. [32], however, found no morphological support for subspecies recognition.

A new BIN (BOLD:AEE3670) was also discovered in *Agriopis bajaria*, constituted by a sample from North-Eastern Austria. The remaining four specimens of this species were collected

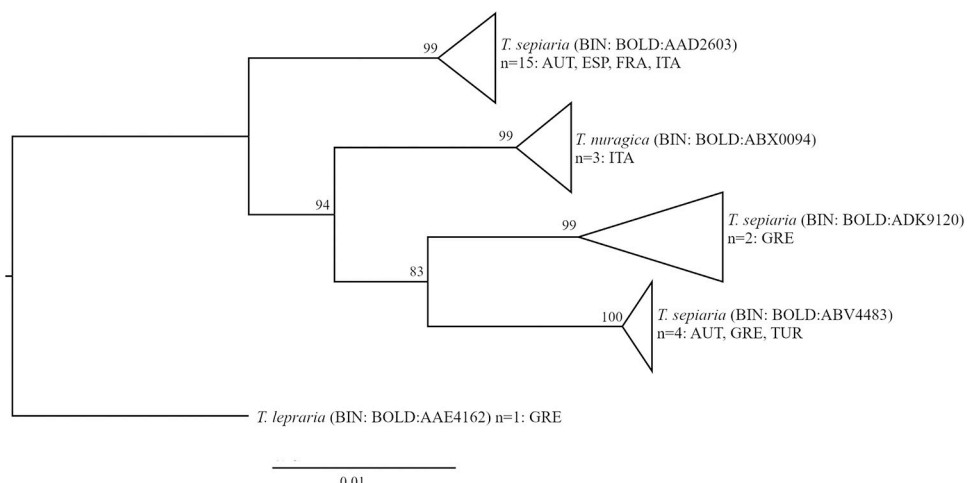

**Fig 8. Phylogenetic relationships among BINs of *Tephronia sepiaria, T. nuragica* and *T. lepraria*.** Neighbor Joining trees based on K2P distances (with pairwise deletion of missing/ambiguous characters), n = sample size; origin of samples indicated with ISO 3166–1 alpha-3 three-letter country code.

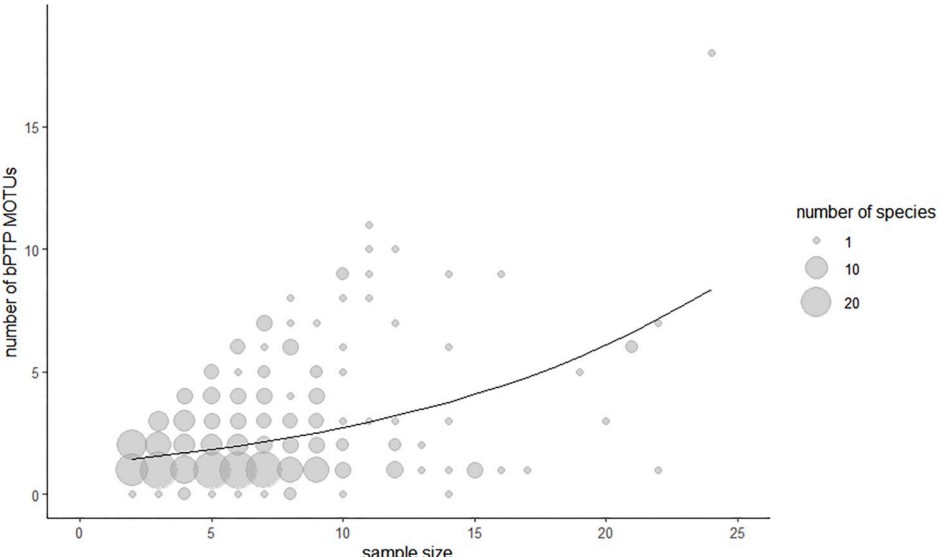

**Fig 9. Increase of bPTP splitting with sample size per species.** For species represented by more than one sample, the number of bPTP MOTUs is plotted against sample size. Dot size represents the number of species with identical values. The line illustrates the relationship predicted by the generalized linear model (predicted number of bPTP MOTUs = exp(0.233539 + sample size * 0.078961)).

in Western and Eastern Austria and share their BIN with specimens from Southern Italy, western Mediterranean, Central-South-Eastern Europe and Lebanon.

In *Tephronia sepiaria*, specimens collected in Western Austria were assigned to a BIN shared by samples from Italy, France and Spain. In contrast, the *T. sepiaria* sample from Eastern Austria belongs to a BIN (BOLD:ABV4483) which has initially been defined based on specimens from central Turkey and for which subspecies status has been taken into consideration [32]. The BIN has then been detected in Greece (Hausmann, unpublished), and is now for the first time reported from Central Europe. A phylogenetic tree based on COI barcode sequences of *T. sepiaria* and its closest relatives *T. nuragica* and *T. lepraria* shows *T. sepiaria* separated in Eastern and Western clades and paraphyletic in relation to *T. nuragica*, which is a species endemic to Corsica and Sardinia (Fig 8). Examination of genitalia morphology in specimens of the two *T. sepiaria* BINs (n = 1 male and 1 female of each BIN) revealed minor differences (Fig 10). Further morphological data need to be collected in order to determine whether these morphological differences are BIN-specific, and the taxonomy of *T. sepiaria* and *T. nuragica* should be further investigated with both genetic and morphological data.

Another interesting taxonomic problem exists in the genus *Crocallis*. The species *C. elinguaria* was found to consist of four BINs (BOLD:AAB0677, BOLD:ACE7622, BOLD: ACF1889, BOLD:AAE4231; [31]), of which BOLD:AAB0677 is widespread across Europe, while BOLD:ACE7622 and BOLD:ACF1889 are found in southern Italy. Twelve Austrian specimens of *C. elinguaria* all assign to BIN BOLD:AAB0677. The last of the four BINs, BOLD: AAE4231, contains specimens from South-Eastern Europe that have been identified as *C. elinguaria*, but the same BIN is shared with a Turkish species, *C. inexpectata*. The taxonomy of the group is currently uncertain, and specimens in this BIN are often referred to as *Crocallis* sp. [31]. Six of the Austrian samples were assigned to this BIN, extending the known distribution of BOLD:AAE4231 to Lower Austria and Carinthia.

The majority of the intraspecific BINs seem to overlap geographically, as their sampling locations do not encompass mutually exclusive areas (Table 3). The classification of

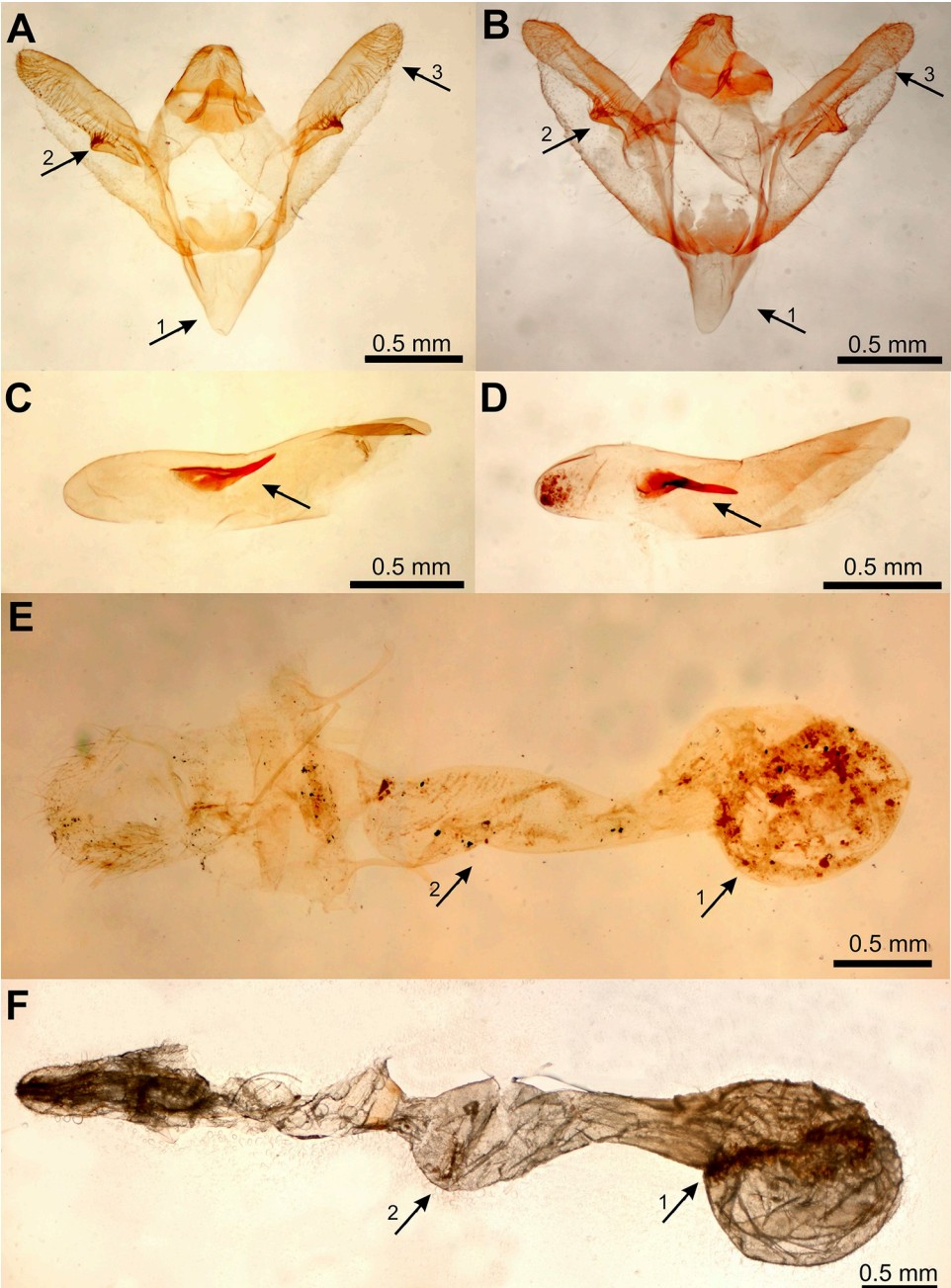

**Fig 10. Genital preparations of *Tephronia sepiaria*.** Genitalia of males in BIN: BOLD:AAD2603 (A) and BIN: BOLD:ABV4483 (B) differ in saccus width (arrow 1), shape of the valva (arrow 2) as well as the number of hairs on and the form of the distinct process on the central part of the valva (arrow 3). Aedeagi of males in BIN: BOLD:AAD2603 (C) and BIN: BOLD:ABV4483 (D) differ in the shape of the cornutus. Genitalia of females in BIN: BOLD:AAD2603 (E) and BIN: BOLD:ABV4483 (F) differ in the form of the signa (arrow 1) and the form of the ductus bursae (arrow 2). Photos by Andreas Eckelt. Specimens: (A, C) Italy, South Tyrol, Naturns, beneath Ladurner; 46.653, 10.975; 650 m; 06.07.2014; leg. P. Huemer. col. TLMF, G 1476 m; sample ID: TLMF Lep 14873. (B, D) Austria, Lower Austria, Bad Vöslau, Harzberg, Steinbruch; 47.9696, 16.1912; 395m; 25.7.2019; leg. C. Wieser; col. KLM, Gig. 12m; sample ID:KLM Lep 08783. (E) Italy, South Tyrol, Montiggl, Kleiner Priol; 46.428, 11.300; 643 m; 30.06.2010; leg. P. Huemer. col. TLMF, G 1478 f; sample ID: TLMF Lep 02391. (F) Austria, Lower Austria, Bad Vöslau, Harzberg, Steinbruch; 47.9696, 16.1912; 395m; 25.7.2019; leg. C. Wieser; col. KLM, GU 21/1529f.

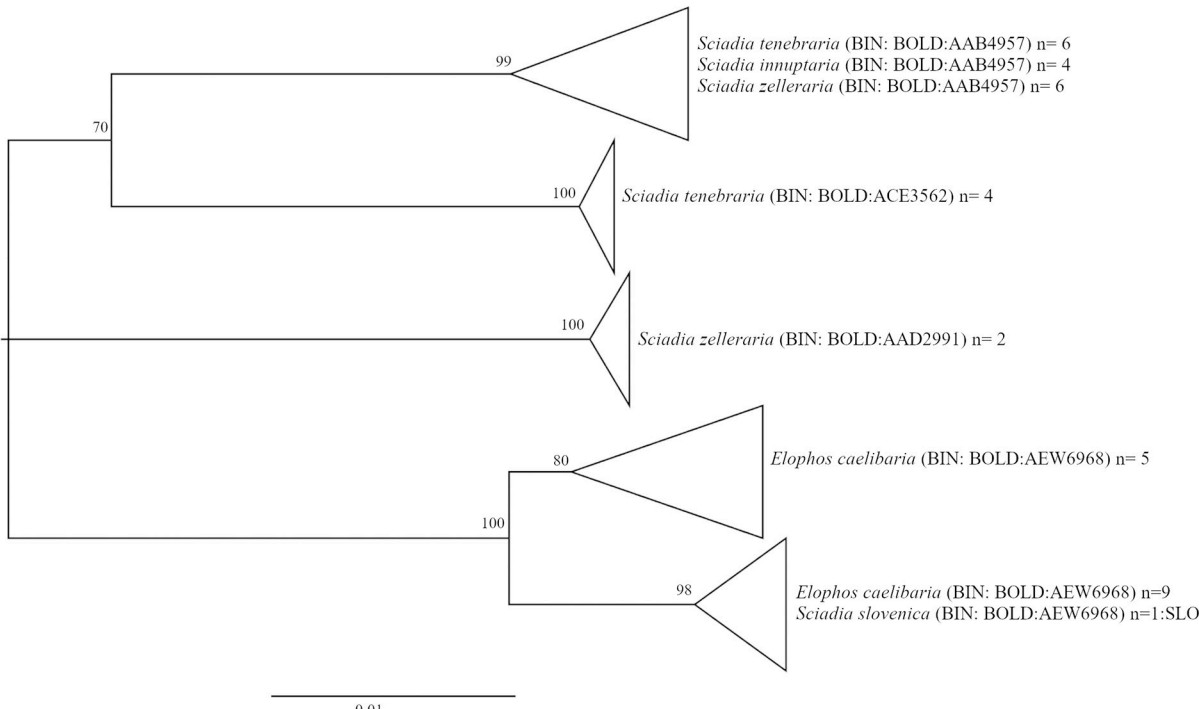

**Fig 11. Phylogenetic relationships among BINs and species of *Sciadia* spp. and *Elophos caelibaria*.** Neighbor Joining trees based on K2P distances (with pairwise deletion of missing/ambiguous characters), n = sample size; samples are of Austrian origin, except for *Sciadia slovenica* (from Slovenia).

'overlapping' versus 'non-overlapping' distributions in Table 3 are based on current BOLD entries, and may prove incorrect when larger sample sizes increase the spatial resolution. For instance, only three DNA barcodes of *Elophos operaria* are available to date, and since the distribution of one BIN (2 samples) does not encompass the location of the other BIN, we scored the two BINs as 'non-overlapping'. Overlapping MOTUs may be reproductively isolated from each other and represent cryptic species; alternatively, divergent mitochondrial lineages may evolve in consequence of Wolbachia infection [84], represent admixture of historically isolated lineages [85] or introgression from other species [86], but can also evolve in large panmictic populations in the absence of population structure [87]. The coincidence of BIN-sharing and BIN-splitting detected in some species of the present study suggests introgression as the source of intraspecific mitochondrial divergence. *Sciadia tenebraria* and *S. zelleraria* are both split into multiple intraspecific BINs, some of which are shared reciprocally and with *S. innuptaria* (Fig 11). Since hybridization of species in the genus *Sciadia* is common [32], introgression of haplotypes is a likely explanation for this BIN pattern. Similarly, Austrian *Rheumaptera subhastata* are split into two BINs, one of which is more closely related to the sister species *R. hastata* than to the conspecific BIN, again a possible consequence of introgression [36]. However, detailed integrative analyses are necessary in each case to elucidate the underlying causes and potential taxonomic implications of the observed BIN splits.

## Conclusions

DNA barcoding and sequence analysis of the complete set of Austrian Geometridae species revealed a high potential for accurate DNA-based species identification, which is promising groundwork for applications with bulk samples or environmental DNA. The present study

also identified 8 species dyads and triads, within which species cannot be distinguished based on COI barcode region sequences (Table 1). Some of them involve morphologically distinct species known to be prone to hybridization. The high level of congruency between morphospecies taxonomy and genetic identification in the present study is, at least in part, due to a history of thorough taxonomic work on the European fauna. We expect that more incongruencies will be detected when studies are extended to less well studied areas [88]. Finally, the table of species investigated in the current study (S1 Table) represents an updated checklist of the geometrid moths of Austria.

## Supporting information

**S1 Table. List of species included in this study, including the number and origin of samples used in the analyses, MOTU delimitation results as well as the minimum distance to, and identity of, the nearest neighbor species.** For each species delimitation method, the resulting MOTUs were given consecutive numbers (IDs of BINs, ASAP-MOTUs and bPTP-MOTUs). MOTUs shared among species are highlighted in yellow.
(XLSX)

**S2 Table. Results of MOTU delimitation analyses, broken down by subfamilies.**
(DOCX)

**S1 Text. Genbank accession numbers.**
(DOCX)

## Acknowledgments

We are particularly grateful to the entire team at the Canadian Centre for DNA Barcoding (Guelph, Canada). We are also grateful to the Ontario Ministry of Research and Innovation and to NSERC for their support of the BOLD informatics platform. We also thank R. Unterasinger for visualising Fig 1 and F. Pühringer, M. Mutanen and G. Fiumi for providing COI sequences. We furthermore acknowledge support, particularly with valuable specimens, from P. Buchner, H. Deutsch, S. Erlacher, S. Erlebach, R. Fauster, P. Gros, A. Haslberger, U. Hiermann, S. Kirchwerger, K. Lechner, A. Mayr, R. Mayrhofer, B. Mueller, A. Ortner, N. Pöll, F. Pühringer, G. Rotheneder, C. Truxer and several additional collectors. We thank Andreas Eckelt for taking photos of the genitalia slides.

## Author Contributions

**Conceptualization:** Benjamin Schattanek-Wiesmair, Peter Huemer, Christian Wieser, Wolfgang Stark, Axel Hausmann, Stephan Koblmüller.

**Data curation:** Benjamin Schattanek-Wiesmair, Peter Huemer, Christian Wieser, Wolfgang Stark, Axel Hausmann.

**Formal analysis:** Benjamin Schattanek-Wiesmair, Stephan Koblmüller, Kristina M. Sefc.

**Funding acquisition:** Peter Huemer, Christian Wieser, Wolfgang Stark, Kristina M. Sefc.

**Investigation:** Benjamin Schattanek-Wiesmair, Peter Huemer, Christian Wieser, Wolfgang Stark.

**Methodology:** Benjamin Schattanek-Wiesmair, Peter Huemer, Stephan Koblmüller, Kristina M. Sefc.

**Project administration:** Benjamin Schattanek-Wiesmair.

**Resources:** Benjamin Schattanek-Wiesmair, Peter Huemer.

**Supervision:** Peter Huemer, Axel Hausmann, Stephan Koblmüller, Kristina M. Sefc.

**Validation:** Benjamin Schattanek-Wiesmair, Peter Huemer, Axel Hausmann, Kristina M. Sefc.

**Visualization:** Benjamin Schattanek-Wiesmair, Kristina M. Sefc.

**Writing – original draft:** Benjamin Schattanek-Wiesmair, Axel Hausmann, Kristina M. Sefc.

**Writing – review & editing:** Benjamin Schattanek-Wiesmair, Peter Huemer, Christian Wieser, Wolfgang Stark, Axel Hausmann, Stephan Koblmüller, Kristina M. Sefc.

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
