## [Decision Letter · Decision Letter 0]

21 Jul 2023

PONE-D-23-17902

A DNA barcode library of Austrian Geometridae (Lepidoptera) reveals high potential for DNA-based species identification

PLOS ONE

Dear Dr. Sefc,

Thank you for submitting your manuscript to PLOS ONE. After careful consideration, we have decided that your manuscript does not meet our criteria for publication and must therefore be rejected.

I am sorry that we cannot be more positive on this occasion, but hope that you appreciate the reasons for this decision.

Kind regards,

Feng ZHANG, Ph.D.

Academic Editor

PLOS ONE

Reviewers' comments:

Reviewer's Responses to Questions

**Comments to the Author**

1. Is the manuscript technically sound, and do the data support the conclusions?

Reviewer #1: Yes

2. Has the statistical analysis been performed appropriately and rigorously? 

Reviewer #1: Yes

3. Have the authors made all data underlying the findings in their manuscript fully available?

Reviewer #1: Yes

4. Is the manuscript presented in an intelligible fashion and written in standard English?

Reviewer #1: No

5. Review Comments to the Author

Reviewer #1: I congratulate the authors established a DNA barcode reference library for all the species of Geometridae that have recorded in Austria, and updated the checklist of the geometrid moths of Austria. The results of this study provides new and comprehensive species diversity resources of Geometridae. However, the writing of the article needs to be improve. The redundancy of sentences is the main problem of this manuscript. I advise the authors to read more articles of DNA barcode to polished the word used in this manuscript. You can also reference the revisions I listed in the attached file. I advise the authors re-submit the manuscript after improve the writing.

6. PLOS authors have the option to publish the peer review history of their article (what does this mean?). If published, this will include your full peer review and any attached files.

Reviewer #1: No

- - - - -

---

## [Author Response · Author response to Decision Letter 0]

11 Aug 2023

Reply to reviewer’s comments.

The manuscript was seen by one reviewer. This reply letter includes the reviewer’s comments followed by our responses.

REVIEWER:

Dear Editor and authors,

I congratulate the authors established a DNA barcode reference library for all the species of Geometridae that have recorded in Austria, and updated the checklist of the

geometrid moths of Austria. The results of this study provides new and comprehensive species diversity resources of Geometridae. However, the writing of the article needs to be improve. The redundancy of sentences is the main problem of this manuscript. I advised the authors to read more articles of DNA barcode to polished the word used in this manuscript. You can also reference the revisions I listed as follows.

REPLY: We thank the reviewer for spotting the replicated sentence (“The assignment of species to subfamilies follows [30].” appeared in line 139 and 175).

Some redundancies in the use of specific terminology are of course inevitable, and could not be removed. Other redundancies were initially intended: Our dataset consists of DNA barcodes generated for this study as well as a small number of barcodes derived from the database in order to complete the assembly. We highlight this both in Material and methods and in the Results section, in order to maintain transparency about the data used in the analyses. Furthermore, we were careful to meticulously and unambiguously report sample sizes used in the different analyses, and this also led to redundances, e.g. when we report whether an analysis included or excluded the data of non-Austrian specimens retrieved from the database. We now removed replicated break-downs of sample sizes in places where we feel that that clarity is nonetheless maintained (L246, L258).

In the beginning of the two sections of the discussion, we summarize what was achieved in this study with regard to the topic of the section (L362-366 in ‘MOTU delimitation and morphospecies identification’; L400-405 and L416-417 in ‘Incongruences between MOTUs and morphospecies’). While this could be considered a redundancy, the summaries are brief, provide a suitable entrance to each section and improve the readability of the manuscript. 

With regard to the use of DNA-barcoding terminology, we would like to note that several of the authors have a strong publication record in DNA-barcoding, totaling over 100 papers, which of course came along with the reading of hundreds of barcoding papers. Additionally, some of the authors have a strong background in phylogenetics and population genetics, again with numerous published papers, and are therefore familiar with the terminology involved in DNA sequence analysis. 

REVIEWER:

Minor comments:

L22: we report a complete DNA barcode library DNA of all the 476 22 species of Geometridae (Lepidoptera) that have been recorded in Austria.

REPLY: The sentence was changed as suggested.

REVIEWER:

L23: Austria. Species were

REPLY: The reviewer suggests to drop “as far as possible” from the beginning of the sentence, which would change the meaning and make the statement slightly incorrect. 

REVIEWER:

L24: capture the intraspecific genetic variation

REPLY: The reviewer suggests to replace “in order to capture intraspecific genetic variation” to “in order to capture the intraspecific genetic variation”. The definite article (“the”) would imply a higher level of completeness in the determination of the amount of genetic variation than achieved in the study. 

REVIEWER:

L25: 2500 DNA barcode sequences

REPLY: Changed as suggested. 

REVIEWER:

L27: with DNA barcodes from

REPLY: Changed as suggested. 

REVIEWER:

L28: by ASAP, BIN and bPTP methods yielded

REPLY: Changed as suggested. 

REVIEWER:

L31:whereas bPTP appeared to overestimate the number of species.

REPLY: The reviewer suggests to replace “whereas bPTP split numerous morphospecies into multiple partitions” with “whereas bPTP appeared to overestimate the number of species”. While we consider the original wording to be more precise, we followed the suggestion for the sake of easy readability. However, we refrain from referring to the bPTP partitions as “species” and therefore write “bPTP appeared to overestimate the number of taxonomic units” (the term MOTU – molecular operational taxonomic unit - is not yet introduced in the abstract).

REVIEWER:

L33-L35: The potential of DNA barcode on species identification has been confirmed over a broad spectrum of biological groups, especially in Lepidoptera. You should clarify that your the DNA barcoding library combining xxx species delimitation method is potential for the species identification of Geometridae (if it is not reported before your study).

REPLY: The statement on the potential for DNA-based species identification indeed refers to the Geometrid species of Austria, and not Lepidoptera in general. We changed the sentence to make this clear (L34): “We conclude that DNA barcoding and sequence analysis revealed a high potential for accurate DNA-based identification of the Austrian Geometridae species”

REVIEWER:

L60: Why do you only compared with the countries of Central European, but not Asia or America? 

REPLY: The paragraph introduces the biodiversity and biogeography of the study region (Austria). Austria is situated in Central Europe, and covers three distinct biogeographic regions. This leads to an exceptionally high level of biodiversity, which stands out in comparison to other Central European countries. In the context of this paragraph, it is therefore meaningful to restrict the comparison to countries within this region. 

REVIEWER:

L61-L63: “species counts have been on the rise in recent decades” and “decline of overall Biomass” , contradiction? 

REPLY: No, this is not a contradiction, because biomass indeed refers to the mass (e.g., in kg) of biological organisms, see for instance Hallmann et al. in PLOS ONE 12(10). Therefore, the species count of a region may rise (due to immigration of new species, or simply due to research progress), but individual numbers (and hence biomass) may decline at the same time. 

REVIEWER:

L69:You should summarize how many species have been obtained the DNA barcode in the world and Austria, respectively. And how many species have not been obtained the DNA barcode.

REPLY: The sentence in question gives examples for barcoding efforts on other Lepidopteran groups, in which Austrian species have been covered to an appreciable extent. A precise account of the current global DNA barcode availability of various Lepidopteran groups (other than the family addressed in this study) does not add useful information to the present study, and the numbers would quickly be outdated. 

REVIEWER:

L70: there is no right parenthesis, please check.

REPLY: Thank you for spotting the missing parenthesis.

REVIEWER:

L81: the taxonomic work of which taxa group?

REPLY: We added “on geometrid moths”.

REVIEWER:

L104: The present study assembled 2500 DNA barcodes of all 476 geometrid species from Austrian, and captured a intraspecific genetic variation based on the broad geographic coverage sampling.

REPLY: the original sentence was “The present study assembles DNA barcodes for all 476 Austrian geometrid species, with a particular effort to sample with broad geographic coverage in order to capture intraspecific genetic variation.“ We followed the reviewer’s suggestion to set the sentence in past tense (and for consistency did the same in the subsequent sentence). As in the comment to L24 (above), we need to avoid the claim that intraspecific variation was fully covered in this study. The sentence was therefore revised as follows (L107): “The present study assembled DNA barcodes for all 476 Austrian geometrid species. A particular effort was made to sample with broad geographic coverage in order to capture intraspecific genetic variation.”

REVIEWER:

L115: we obtained DNA barcode sequences

REPLY: The reviewer suggests to replace “we compiled barcode sequences” by “we obtained barcode sequences”. We suggest a different alternative wording (“We compiled a dataset comprising barcode sequences …”), because it is more precise: While (as explained in the next sentence in the manuscript) we generated barcode sequences for 438 species, our dataset for the 476 Austrian geometrid species also included barcode sequences for 38 species, which were taken from the BOLD database.

REVIEWER:

L116: Herein, 438 species

REPLY: the reviewer suggests to replace “Of these, 438 species …” by “Herein, 438 species …”. We feel that the original wording is standard and better understandable. 

REVIEWER:

L116:at least one specimen

REPLY: Changed as suggested. 

REVIEWER:

L117:by the specimens from non-Austrian

REPLY: The original wording was “sample of non-Austrian origin“, and we replaced this by “specimens of non-Austrian origin“.

REVIEWER:

L119: to this study

REPLY: Changed as suggested. 

REVIEWER:

L119-L120: study, most specimens are originating from Tiroler

REPLY: Changed as suggested. 

REVIEWER:

L124: We divided Austria in three

REPLY: the reviewer suggested to drop “therefore” from the sentence. Since the sampling strategy followed from the intention explained in the previous sentence, we suggest to keep the therefore, but changed the word order: “Therefore, we divided Austria in three areas”.

REVIEWER:

L131: the three areas

REPLY: Changed as suggested. 

REVIEWER:

L132-L133: Do you mean sample at least three specimens per species in each defined area? Why the species were known to occur in only one or two of the defined areas could be sampled at least three specimens per species, the species were known to occur in three area should not?

REPLY: We regret that we do not fully understand the question, but we revised the text to make the sampling design clear (L133): “Sample coverage of the three areas was achieved for 272 species, and 79 species were sampled from two areas. The remaining species either occur in only one of the areas [4], or sample collection from the other areas failed although the species has been recorded there. In any case, we aimed at sampling at least three specimens per species.”. 

REVIEWER:

L135-L138: we retrieved 38 COI sequences of these 38 species from the BOLD database, and their sampling sites are near Austria (mostly from Germany......) 

REPLY: the original sentence read “In order to represent these species in the analyses, we retrieved one COI sequence per species from the BOLD database, using the geographically nearest sampling sites (mostly from Germany but ranging from the Iberian Peninsula to Finland; S1 Table).”

The suggested wording would imply that we used 38 sequences per species, which was not the case. The reviewer also appears to object to the wording in “using the geographically nearest sampling sites”, and we resolved this as follows (L 140): “… we retrieved one COI sequence per species from the Barcode of Life Data System (BOLD, www.boldsystems.org; [52]), selecting database entries from the geographically nearest sampling sites…”

REVIEWER:

L151: sequencing the DNA barcode region

REPLY: Changed as suggested. 

REVIEWER:

L152:The DNA barcode sequences

REPLY: Changed as suggested. 

REVIEWER:

L153:generated based on a standard high-throughput protocol [53] with primers

REPLY: The original wording “sequences were generated using a standard high-throughput protocol” is standard in protocol reporting; however, we changed “using primers” to “with primers”, as suggested. 

REVIEWER:

L154-156: LepF2 [54]. The length of sequences met the criterias of BOLD system (Barcode of Life Data System, www.boldsystems.org[55]; i.e., at least 500 bp) were retained for the subsequent analyses.

‘BOLD’ should be show as full name for the first appearance, and next time just show BOLD is ok. Please check that throughout the manuscript

REPLY: We changed this as suggested. BOLD is given with full name, URL and reference in line 141, and thereafter referred to as “BOLD” only. The only exception is line 171, where we specify the version; this is appropriate in this sentence, as at describes the calculations done within the BOLD system. 

REVIEWER:

L156: specimen collection data and

REPLY: Changed as suggested. 

REVIEWER:

L157-158: Austria” in the BOLD, and DNA sequences were available in

L158-159:DNA sequences were also deposited in Genbank (Genbank accession numbers XXX -XXX)

REPLY: The two comments refer to the same sentence, which was changed to “Specimen collection data and images are publicly available in the BOLD dataset “DS-LEATGEOM Lepidoptera (Geometridae) of Austria” , and DNA sequences were also deposited in Genbank (Genbank accession numbers XXX -XXX).” (L161)

REVIEWER:

L162: For each species, The the nearest neighbor distance and the maximum intraspecific distance were calculated by the Barcode Gap Analysis tool of the BOLD system, based on the Kimura two-parameter (K2P) model.

The meaning of “the nearest” is same to “ the minimum” for nearest neighbor distance. 

REPLY: The original wording is: “the minimum Kimura two-parameter (K2P) distance to the nearest neighbor species in the dataset (min.NN)“, and distinguishes between the most closely related species (“nearest neighbor species”) and the DNA sequence (among the sequences available from the nearest neighbor species) that is most similar to the sequences of the focal species. In other words, the database may contain several barcoding sequences from the most closely related species, and “min.NN” is the genetic distance to the most similar of these sequences. In order to clarify, we rephrased as follows (L168): “For each species, the nearest neighbor Kimura two-parameter (K2P) distances (min.NN) as well as mean and maximum intraspecific K2P distances (for species with > 1 sample) were calculated within the BOLD system v. 4.0 using the Barcode Gap Analysis tool, with pairwise deletion of missing/ambiguous characters. Nearest neighbor distances were calculated between the focal species and the most similar COI sequence of the nearest neighbor species in the dataset.”

REVIEWER:

L168-170: taxonomic assignment. The species with BIN split were checked for the morphological identification. These data

REPLY: The reviewer suggests to replace the sentence “We also identified those of the Austrian species, which are split into multiple BINs based on datasets that include all European specimens (excluding Russian and Turkish specimens)” with “The species with BIN split were checked for the morphological identification”. The suggested sentence does not convey the intended meaning. We agree that the original sentence was hard to understand and apologize for this. We revised this section (L177): “We recorded the number of BINs per species, that were detected in the Austrian specimens. For each of the 476 Austrian geometrid species, we also determined the number of BINs per species in datasets including all European specimens (excluding Russian and Turkish specimens).”

REVIEWER:

L175: “The assignment of species to subfamilies follows [30].” is redundancy, you have mentioned in L139. And the L139 should be polished.

REPLY: This sentence was duplicated, and has been removed from L139. In L 186, we revised the sentence: “The assignment of species to subfamilies follows the systematic checklist of European Geometridae [30].”

---

## [Decision Letter · Decision Letter 1]

15 Nov 2023

PONE-D-23-17902R1A DNA barcode library of Austrian Geometridae (Lepidoptera) reveals high potential for DNA-based species identificationPLOS ONE

Dear Dr. Kristina Sefc,

Thank you for submitting your manuscript to PLOS ONE. After careful consideration, we feel that it has merit but does not fully meet PLOS ONE’s publication criteria as it currently stands. Therefore, we invite you to submit a revised version of the manuscript that addresses the points raised during the review process.

We look forward to receiving your revised manuscript.

Kind regards,

Feng ZHANG, Ph.D.

Academic Editor

PLOS ONE

Journal Requirements:

Funding for DNA sequencing was provided by the following institutions:

- Landesmuseum Kärnten CW, https://landesmuseum.ktn.gv.at/

- Landessammlungen Niederösterreich WS, https://www.landessammlungen-noe.at

- Tiroler Landesmuseen PH, BSW https://www.tiroler-landesmuseen.at/

- Promotion of Educational Policies, University and Research Department of the Autonomous Province of Bolzano - South Tyrol PH, https://errin.eu/members/autonomous-province-bolzanobozen-south-tyrol

Funding for Open Access publication was provided by University of Graz.

4. We note that Figure 1 in your submission contain map images which may be copyrighted. All PLOS content is published under the Creative Commons Attribution License (CC BY 4.0), which means that the manuscript, images, and Supporting Information files will be freely available online, and any third party is permitted to access, download, copy, distribute, and use these materials in any way, even commercially, with proper attribution. For these reasons, we cannot publish previously copyrighted maps or satellite images created using proprietary data, such as Google software (Google Maps, Street View, and Earth). For more information, see our copyright guidelines: http://journals.plos.org/plosone/s/licenses-and-copyright.

Additional Editor Comments (if provided):

Reviewers' comments:

Reviewer's Responses to Questions

**Comments to the Author**

1. If the authors have adequately addressed your comments raised in a previous round of review and you feel that this manuscript is now acceptable for publication, you may indicate that here to bypass the “Comments to the Author” section, enter your conflict of interest statement in the “Confidential to Editor” section, and submit your "Accept" recommendation.

Reviewer #1: All comments have been addressed

2. Is the manuscript technically sound, and do the data support the conclusions?

Reviewer #1: Yes

3. Has the statistical analysis been performed appropriately and rigorously? 

Reviewer #1: Yes

4. Have the authors made all data underlying the findings in their manuscript fully available?

Reviewer #1: Yes

5. Is the manuscript presented in an intelligible fashion and written in standard English?

Reviewer #1: Yes

6. Review Comments to the Author

Reviewer #1: (No Response)

7. PLOS authors have the option to publish the peer review history of their article (what does this mean?). If published, this will include your full peer review and any attached files.

Reviewer #1: No

---

## [Author Response · Author response to Decision Letter 1]

2 Jan 2024

Dear Editors, 

In the following, we respond to each of the referee’s comments and explain the changes made in the manuscript. Reviewer comments are in bold print (in uploaded file). 

Comments to the Author

The authors made efforts in improving the manuscript. And a few revisions are required. I congratulate the authors for this ambitious study. Revisions are as follows:

L25: In total, 2500 DNA barcode sequences (“DNA barcode” can be more precise, “barcode” could be barcode of goods. Some professors who study in DNA barcode have given me that suggestion. So please check that throughout the manuscript) 

REPLY: We checked all occurrences of “barcode” and replaced them with “DNA barcode”. 

L137: “sites(mostly from” change to “sites (mostly from”

REPLY: we inserted the space

L162: calculated on the BOLD system

REPLY: changed as suggested by the reviewer

L184: MAFFT, Modelfinder and IQ-TREE should be clarified the version number, please check all software throughout the manuscript.

REPLY: we added version number to all software, provided that version numbers are attached to the software (this was not the case for ASAP and bPTP)

L204: “and 2525 sequences were assigned a Barcode Index Number (BIN)” Do you mean “unique” BIN? If not, why are the remaining 42 sequences not assigned BIN? Please clarify.

REPLY: No, we do not mean “unique BIN”; these are sequences that were assigned a BIN number, which could be shared with other sequences. There are several possible reasons for why some (i.e., 42) sequences were not assigned a BIN at the time of data analysis – for instance that sequences were too short, references flagged as false identification, or only recently uploaded. 

L220: 7.1 % of xxx. It is ambiguous. Please clarify.

REPLY: was changed to “7.1 % of 476 species”

L246: 27 morphospecies presented sharing BINs.(Your original sentence could be mean “27 morphospecies, they shared BINs with each other”)

REPLY: we changed the wording to “27 morphospecies shared BINs with another morphospecies”

L283: , and two BINs were presented in each of 47 species.

REPLY: We changed the wording to “In Austria, 91 of these species are represented with a single BIN each, and 47 species are each represented with two BINs (Fig 6; S1 Table)”.

L309: Based on the ASAP method

REPLY: changed as suggested by the reviewer

L364-365: ASAP performed similarly well in morphological taxonomy, it is actually related to the DNA barcode gap (max. Intraspec and min.NN ), it would be better if you discuss about that. 

REPLY: we revised the sentence: “In the present study, BIN and ASAP performed similarly well in terms of congruence between reconstructed MOTUs and morphological taxonomy”. 

All three algorithms seek to identify the point, where intraspecific diversity can be distinguished from interspecific diversity, so the barcode gap is relevant for each of the methods. 

L400: Please check that Fig 7 is right (the Fig number)

REPLY: the figure number was corrected.

L430-432: Please show the relevant figure of genitalia.

REPLY: we include photographs of genitalia of male and female 

L464: cf. should be followed by the species name. Please clarify.

REPLY: the “cf” was a typo and deleted

For Table 1: as you show in Table 1, most of species sharing BINs show differentiation in wing coloration and wing shape. If that suggested the wing coloration and wing shape may be not suitable for species identification? If there are some controversy on using the wing coloration and wing shape to identify species? You should discuss that in Discussion.

REPLY: When BIN sharing species differ in morphological traits, BIN sharing can be a consequence of introgression, i.e. hybridization followed by introduction of one species’ mt DNA haplotype into the other species. These findings do not raise a controversy particularly on the use of wing coloration or wing shape for species identification, as the phenomenon can concern any morphological trait. 

L278: Fig5: You should clarify the tree you have showed, for example, NJ tree or ML tree. The figure and its legend must be understandable without reference to the text.

REPLY: we added the method (Neighbor joining) to the figure legend

And the species name in Fig5 should be italic.

Please check the Fig7, Fig8, and Fig10, accordingly.

REPLY: changed as requested in all figure legends

For Fig9: Please show the regression equation, R-value and P-value of the curve in the figure.

REPLY: The line in the graph shows the predictions of a negative binomial regression model. The relationship between the predictor and the response variable in the regression equation is logarithmic. Therefore, the regression equation cannot easily be interpreted and would not add useful information to the graph. To comply with the reviewer’s request to report the equation, we included it in the figure description. We do not repeat the p-value in the figure or its description, as it is reported in the text together with the effect size estimate (effect size, test statistic and p-value are reported as est. = 0.08, z = 8.78, p < 10-16).

---

## [Editor Report · Decision Letter 2]

17 Jan 2024

A DNA barcode library of Austrian Geometridae (Lepidoptera) reveals high potential for DNA-based species identification

PONE-D-23-17902R2

Dear Dr. Kristina Sefc,

We’re pleased to inform you that your manuscript has been judged scientifically suitable for publication and will be formally accepted for publication once it meets all outstanding technical requirements.

Kind regards,

Feng ZHANG, Ph.D.

Academic Editor

PLOS ONE
---

## [Editor Report · Acceptance letter]

27 Feb 2024

PONE-D-23-17902R2 

PLOS ONE

Dear Dr. Sefc, 

I'm pleased to inform you that your manuscript has been deemed suitable for publication in PLOS ONE. Congratulations! Your manuscript is now being handed over to our production team.

Kind regards, 

on behalf of

Dr. Feng ZHANG 

Academic Editor

PLOS ONE